# Epidemiology and demographics of slipped capital femoral epiphysis in South Korea: A nationwide cohort study

**Nak Tscheol Kim[1☯], Jae Jung Min[1☯], Eunjeong Ji[2], Moon Seok Park[1], Ki Hyuk Sung[1]\***

**1** Department of Orthopedic Surgery, Seoul National University College of Medicine, Seoul National University Bundang Hospital, Seongnam, Gyeonggi, Korea, **2** Medical Research Collaborating Center, Seoul National University Bundang Hospital, Seongnam, Korea

☯ These authors contributed equally to this work.
\* skh1219@naver.com

**Data Availability Statement:** The data underlying the results presented in the study are available from Korean Health Insurance Review and

## Abstract

### Background

This study investigated the epidemiology and demographics of slipped capital femoral epiphysis (SCFE) in South Korea using a nationwide population-based database.

### Methods

Information on sex, age at onset, endocrine comorbidities, history of growth hormone therapy, history of radiation therapy, surgical methods, and complications in patients with SCFE aged younger than 18 years between 2007 and 2019 was retrieved from the Korean Health Insurance Review and Assessment Service database.

### Results

Data were available for 586 children (429 boys, 157 girls). The average age at onset was 11.1 ± 1.8 years (boys, 11.3 ± 1.9; girls, 10.6 ± 1.5). Five hundred and twenty-nine (90.3%) patients were aged 9–14 years; the incidence rate in this age group was 6.0/100,000 (95% confidence interval, 5.5–6.6) and significantly higher in boys (8.4 vs 3.5, p<0.001). There was a significant increase in the annual incidence rate from 0.96/100,000 in 2009 to 2.05/100,000 in 2019 (p = 0.006). Ninety-five patients (16.2%) had endocrine comorbidities or a history of hormone/radiation therapy. Osteotomy and internal fixation were performed in 59 patients and in situ pinning in 527. Seventy-six patients developed postoperative complications.

### Conclusion

The annual incidence rate of SCFE in South Korea has increased since 2009.

Assessment Service (HIRA) database. https://opendata.hira.or.kr/home.do.

**Funding:** This work was supported by the National Research Foundation of Korea(NRF) grant funded by the Korea government(MSIT) (No. NRF-2019R1C1C1010352) and by the Korea Medical Device Development Fund grant funded by the Korea government (Project Number: RS-2020-KD000308).

**Competing interests:** The authors have declared that no competing interests exist.

## Introduction

Slipped capital femoral epiphysis (SCFE) is associated with many complications and deformities secondary to a shift in position of the epiphysis in relation to the metaphysis of the proximal femur [1]. Possible sequelae of SCFE include avascular necrosis, chondrolysis, leg length discrepancy (LLD), and early degenerative joint disease [2, 3]. Obesity, endocrine, and systemic disorders are known to be associated with an increased incidence of SCFE [4–7]. When combined with increased body weight and retroversion of the femoral head, the loads may exceed the ultimate yield point, causing a shift in position of the epiphysis in relation to the metaphysis of the femoral head. In severe cases, which are classified as unstable, a true separation may occur [8].

Reports of the prevalence of SCFE vary from country to country. SCFE is a common hip disorder in children aged 9–15 years in Western countries and is the most common reason for hip replacement surgery in adolescents but is thought to be relatively rare in Korea [9–11]. Since the first published report of SCFE in Korea in 1979, the incidence of the disorder has been increasing. A multicenter study of the epidemiology of SCFE in Korea between 1989 and 2003 showed a continuous increment in its incidence [11]. Given the increase in the number of reported cases of SCFE since then, a more up-to-date study on the incidence of SCFE seemed warranted.

In this study, we investigated the epidemiology and demographics of patients with SCFE in South Korea using a nationwide population-based database based on the hypothesis that the incidence rate of SCFE would be increasing year by year.

## Methods

The study protocol was approved by the institutional review board of Seoul National University Bundang Hospital (IRB No.X-2006/621-904). The need for informed consent was waived in view of the retrospective observational design of the study. Data for patients under 18 years of age who were diagnosed to have SCFE and underwent related surgery within 90 days between 2007 and 2019 were retrieved from the Korean Health Insurance Review and Assessment Service (HIRA) database. All Korean patients are enlisted in HIRA, as the Korean healthcare is a national health insurance. Nearly all information on patients and their medical records can be obtained from this database, which has been used in previous epidemiology studies. Accuracy of the database regarding SCFE is guaranteed as patients diagnosed with SCFE are registered in HIRA upon diagnosis made by pediatric orthopedic specialists [12–14]. The washout period was set at 2 years to exclude patients in whom the onset of SCFE was before the study period. The data extracted included diagnosis, sex, age at onset, endocrine diseases, history of growth hormone therapy, history of radiation therapy, surgical methods, and complications.

The incidence of SCFE was expressed as new cases per 100,000 children and calculated based on data for the population in the age group at highest risk (9–14 years) between 2009 and 2019, which were obtained from the Korean Statistical Information Service [15].

Complications after surgery for SCFE were defined as avascular necrosis, femoroacetabular impingement, chondrolysis, LLD, and genu valgum. Avascular necrosis, femoroacetabular impingement, and chondrolysis were identified by their diagnostic codes alone. LLD and genu valgum were considered to have occurred based on both diagnostic and surgical codes (epiphysiodesis or hemiepiphysiodesis).

### Statistical analysis

The Mann-Kendall test and the Sen's slope estimator were used to determine the significance of trends in our data. The study data were analyzed using SAS Enterprise Guide version 6.1

(SAS Institute Inc., Cary, NC, USA) and R Statistical Software version 3.5.2 (R Foundation for Statistical Computing, Vienna, Austria). A p-value <0.05 was considered statistically significant.

## Results

Data were retrieved for 586 patients with SCFE (429 boys, 157 girls; ratio 2.73:1). The average age at onset was 11.1 ± 1.8 years (11.3 ± 1.9 years in boys; 10.6 ± 1.5 years in girls). The median age at onset was 12 years for boys (range, 5–18 years) and 11 years for girls (range, 6–18 years). Five hundred and twenty-nine (90.3%) of the 586 patients were aged 9–14 years and comprised 382 boys and 147 girls.

The incidence rate in the children aged 9–14 years was 6.0 per 100,000 (95% confidence interval, 5.5–6.6) and was significantly higher in boys than in girls (8.4 vs 3.5, p<0.001). The highest annual incidence rate occurred at the age of 12 years in boys (2.4 cases per 100,000) and at 11 years in girls (1.2 cases per 100,000; Fig 1).

The annual incidence rate in the at-risk population increased significantly from 0.96 per 100,000 in 2009 to 2.05 in 2019 (p = 0.006; Fig 2). The rate in boys increased significantly from 1.06 to 2.38 between 2009 and 2019 (p = 0.006). There was also a significant increase in the rate for girls from 0.58 in 2009 to 1.19 in 2019 (p = 0.024).

Ninety-five (16.2%) of the 586 patients were found to have endocrine comorbidities, a history of hormone therapy, or to have received radiation therapy. Thyroid disease (60 patients,

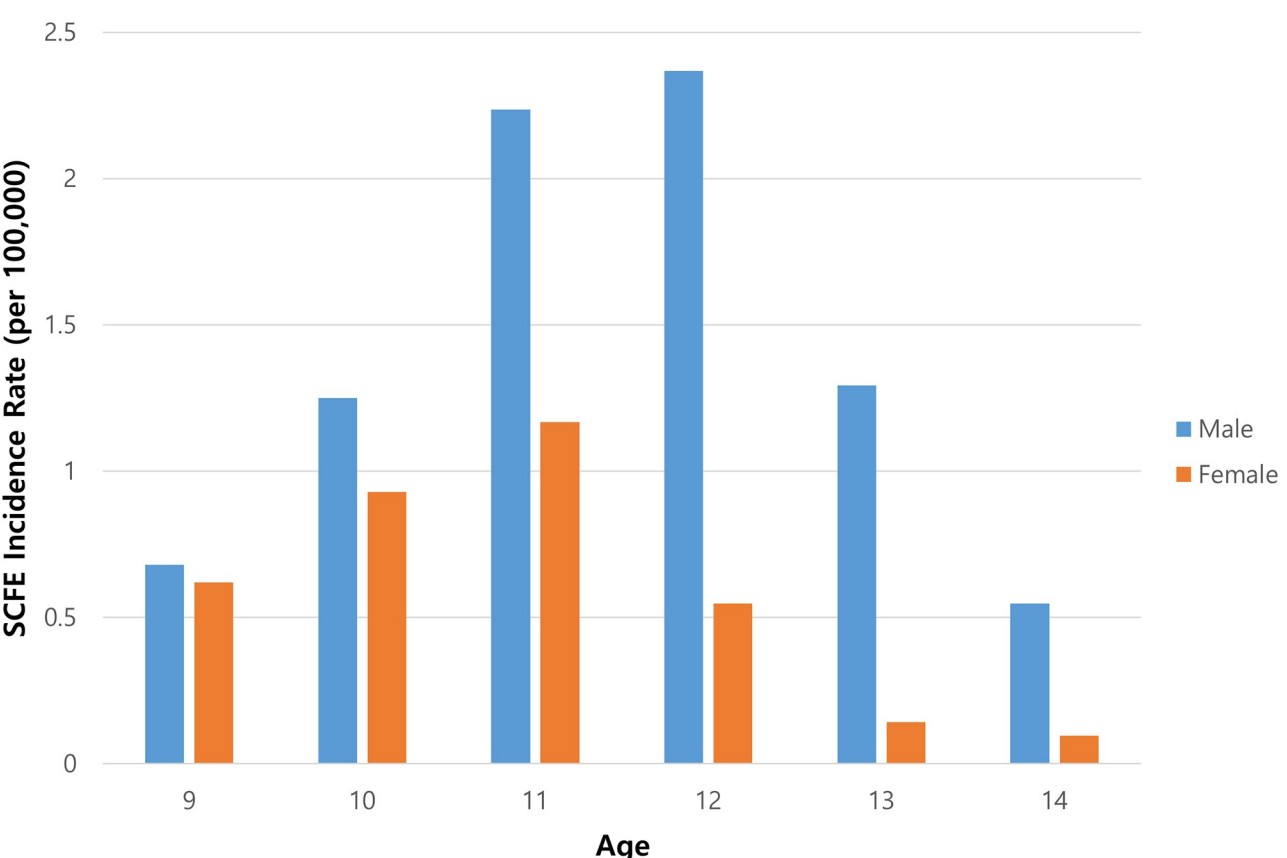

**Fig 1. Incidence rate of slipped capital femoral epiphysis by age group.**

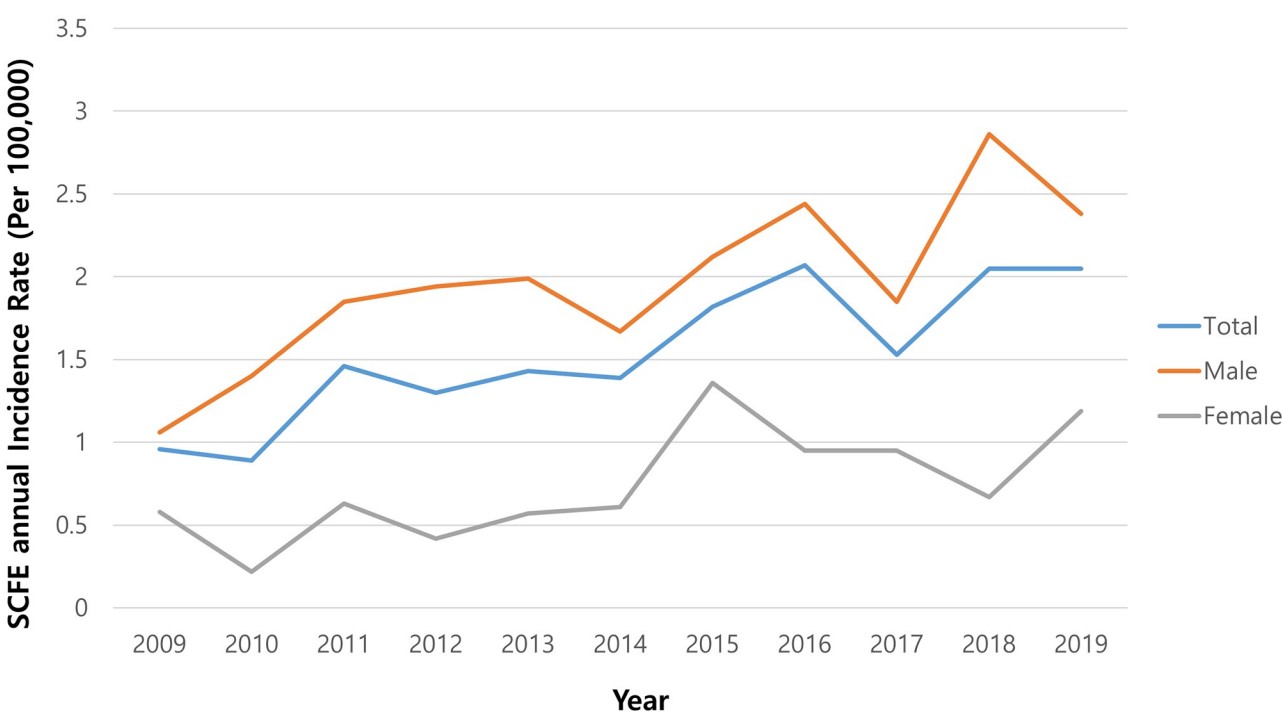

**Fig 2. Annual incidence rate for slipped capital femoral epiphysis in children aged 9–14 years.**

63%) was the most common comorbidity followed by pituitary disease (29 patients, 30%; Table 1).

Fifty-nine patients underwent osteotomy and internal fixation and 527 underwent in situ pinning. Bilateral surgery was performed in 79 patients (13.5%) and involved in situ pinning in all cases. Bilateral hip surgery was performed on the same day in 15 cases and contralateral surgery at an average of 13.6 months after the initial surgery in the remaining 64 (10.9%).

Avascular necrosis occurred at a mean of 26.4 months after the initial surgery for SCFE in 25 patients. Twelve patients with AVN have undergone femoral osteotomy, with one case each of pelvic osteotomy and total hip arthroplasty. Femoroacetabular impingement (pistol grip deformity) occurred at an average of 45.6 months after the initial surgery in 13 patients and chondrolysis at an average of 9.3 months in two patients. No additional surgery was performed

**Table 1. Diseases and history of hormone and radiation therapy in SCFE patients.**

|  | Total SCFE patients (N = 586) | MALE (N = 429) | FEMALE (N = 157) |
|---|---|---|---|
| **Diseases** |  |  |  |
| Pituitary gland disease | 29 (4.95) | 11 (2.56) | 18 (11.46) |
| Thyroid disease | 60 (10.24) | 30 (6.99) | 30 (19.11) |
| Parathyoid disease | 12 (2.05) | 8 (1.86) | 4 (2.55) |
| Hypogonadism | 16 (2.73) | 12 (2.80) | 4 (2.55) |
| Turner syndrome | 1 (0.17) | 0 (0.00) | 1 (0.64) |
| Scurvy | 1 (0.17) | 1 (0.23) | 0 (0.00) |
| History of hormone therapy |  |  |  |
| Growth hormone | 10 (1.71) | 4 (0.93) | 6 (3.82) |
| History of radiation therapy | 11 (1.88) | 5 (1.17) | 6 (3.82) |

**Table 2. Complications after SCFE surgery.**

| Complication | Patients number (%) | Mean onset (months) |
|---|---|---|
| Avascular necrosis | 25 (4.27) | 26.4 |
| Femoroacetabular impingement | 13 (2.22) | 45.6 |
| chondrolysis | 2 (0.34) | 9.3 |
| Leg length inequality | 16 (2.73) | 23.8 |
| Genu valgum | 20 (3.41) | 31.3 |

in these cases. Sixteen patients underwent epiphysiodesis for LLD at an average of 23.8 months after the initial surgery and 20 underwent hemiepiphysiodesis for genu valgum at an average of 31.3 months (Table 2).

## Discussion

There are wide geographic differences in the incidence rate of SCFE. The overall incidence rate has been reported to be 10.8 per 100,000 in the US and 81 per 100,000 annually in the Maori population of New Zealand [10, 16]. In the Netherlands, the incidence of surgical procedures for SCFE has been reported to be 11.6 per 100,000 children aged 5–19 years [17]. The average annual incidence in Sweden was reported to be 4.4 per 10,000 in girls and 5.7 per 10,000 in boys aged 9–15 years [1]. In Asia, the incidence rate is markedly lower than in Western countries [11, 18–20]. The average annual incidence in Japan was estimated to be 2.22 for boys and 0.76 for girls per 100,000 in the population aged 10–14 years between 1997 and 1999, which is five times higher than the statistics from the eastern half of Japan in 1976 [18] (Table 3).

The reasons for these marked regional differences in incidence are not clear but may reflect genetic and environmental differences between the various populations. Several studies have indicated that genetics are involved in the etiology of SCFE [21, 22].

Like in Japan, the incidence of SCFE in South Korea is significantly lower than that in the West. However, in this study, there was a 113% increase in the incidence rate (from 0.96 per 100,000 in 2009 to 2.05 in 2019). Between 1989 and 2003, the estimated average annual incidence of SCFE per 100,000 children in the at-risk population was 0.328 in Korea [11]. This steady increase in incidence in a population that is racially homogeneous suggests that SCFE is influenced not only by genetic factors but also by environmental factors.

One of the environmental factors affecting the incidence of SCFE is obesity [23]. Rates of stress, strain, and displacement of the physis have been reported to increase with increasing body weight, and mechanical factors such as obesity are known to be etiologically more important than endocrine abnormalities [8, 24]. Increased shear stress due to obesity is likely to be the most critical factor. We could not test for a direct correlation between the incidence of SCFE and obesity because BMI was not included in the HIRA data. However, the obesity rate among Korean children and adolescents is steadily increasing (Fig 3) [25]. We assume that this steady increase in the rate of obesity, which is already a known risk factor for SCFE, may be an important environmental factor influencing the increased incidence of SCFE in Korea. In the clinical setting, SCFE should be included in the differential diagnosis of adolescent patients with hip pain.

It is generally known that boys are more susceptible to SCFE, although there is considerable variation worldwide. In Sweden, SCFE has a reported male predominance of about 60% [26]. Loder et al. reviewed 1630 children with 1993 SCFE and reported that 41.2% were girls and 58.8% were boys [9]. Their study population included white, black, Amerindian, Indonesian-

**Table 3. Summary of SCFE incidence as noted in previous literature.**

| References | Country | Database | Duration | Patient age (yrs) | SCFE incidence | Trend | Male-to-female ratio |
|---|---|---|---|---|---|---|---|
| Song et al. [11] | Korea | Multicenter study | Jan 1989-Dec 2003 | 10–14 | 0.3/100,000 | Increasing | 3.5:1 |
| Perry et al. [29] | UK | NHS | Jan 1990-Mar 2014 | 0–16 | 4.8/100,000 | Constant | 1.7:1 |
| Noguchi et al. [18] | Japan | Multicenter study | Jan 1997-Dec 1999 | 0–14 | 2.2/100,000 | Increasing | 5:1 |
| Lehmann et al. [10] | USA | 1997, 2000 KID# national data | 1997, 2000 | 9–16 | 10.8/100,000 | Increasing | 1.7:1 |
| Witbreuk et al. [17] | The Netherlands | NHS | 1998–2010 | 5–19 | 11.6/100,000 | Increase female population | - |
| Phadnis et al. [16] | Maori | Regional institute | 2000–2010 | 5–14 | 81/100,000 | n/a | 2:1 |
| Ravinsky et al. [30] | Ontario, Canada | ICES** data | Apr 2002-Mar 2011 | 0–16 | 5.7/100,000 | Decreasing | n/a |
| Fedorak et al. [32] | Samoa | Regional institute | 2005–2014 | 5–14 | 53.1/100,000 | Increasing | 2:1 |
| Herngren et al. [1] | Sweden | NHS* | 2007–2013 | 9–15 | 4.4/10,000 girls  5.7/10,000 boys | Increasing in female population | 1.3:1 |
| Tucker et al. [31] | Northern Ireland | PACS## | Jan 2013-Dec 2015 | <16 | 4.7/100,000 | Decreasing | 2:1 |
| Current study | Korea | K-HIRA data | 2007–2019 | <18 | 6/100,000 | Increasing | 2.4:1 |

NHS, National Health Service; KID, Kids' Inpatient Database; ICES, Institute for Clinical and Evaluative Sciences; PACS, Picture archiving and communication system; K-HIRA, Korean Health Insurance Review and Assessment Service

Malay, Native Australian/Pacific Island, and Indo-Mediterranean children. By contrast, no statistically significant sex-related difference in the incidence of SCFE was found in the Netherlands [17]. In our present study, the incidence of SCFE was 2.4 times higher in boys than in girls (8.4 vs 3.5) and the difference was statistically significant.

Swarup et al. reviewed 9755 patients in the US who had undergone in situ pinning for unilateral SCFE and found that 1077 (11%) subsequently developed SCFE on the contralateral side after an average of 277 days [27]. In a Korean study by Song et al., 19% of patients were reported to have bilateral slippage [11]. Chan et al. found that SCFE developed on the contralateral side in 7 (16.3%) of 43 patients with SCFE and that the mean interval between the index surgery and contralateral fixation was 43.9 weeks [28]. In our study, 79 patients (13.5%) underwent bilateral hip surgery. In 15 cases, bilateral surgery was performed on the same day; however, whether these procedures were done for prophylactic purposes or for treatment of bilateral SCFE cannot be ascertained given the nature of the HIRA database. In the remaining 64 patients (10.9%), surgery for contralateral SCFE was performed after an average of 13.6 months. Although controversy persists regarding the value of prophylactic pinning of the unaffected side after unilateral SCFE, we believe that patients should be followed until skeletal maturity at least to monitor for contralateral SCFE.

This study has some limitations, which stem mainly from its retrospective cohort design. First, we could not account for misclassifications, such as incorrect billing or wrong diagnostic codes. However, only patients with a diagnosis of SCFE who underwent related surgery within 90 days were included in the study, which we believe minimized the likelihood of the initial

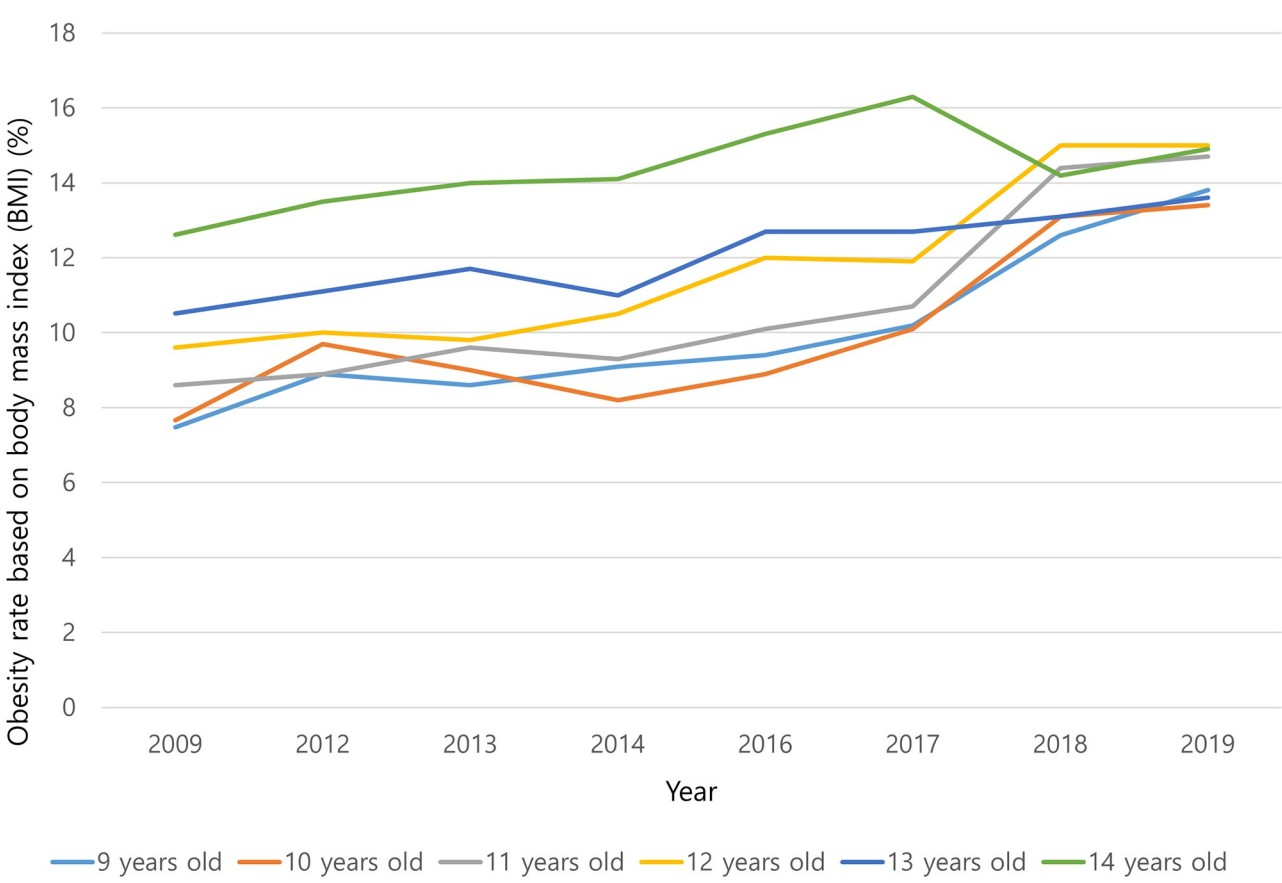

**Fig 3. Annual increase in obesity rate based on body mass index (BMI) since 2009 in the Korean population aged 9–14 years.** *BMI was assessed by age-specific and sex-specific percentiles and obesity was defined as a BMI above the 95th percentile.

diagnosis being incorrect. Second, we could not test for a correlation between the incidence of SCFE and obesity because the HIRA database does not include information on BMI. Third, it is difficult to discern the reason for simultaneous bilateral hip surgeries with the HIRA database. We cannot know whether simultaneous bilateral hip surgeries were done for prophylactic purposes or for treatment of bilateral SCFE, for the database lacks information on laterality. Fourth, the type of SCFE cannot be distinguished with the HIRA database. Therefore, the rate of stable and unstable SCFE, as well as whether AVN after SCFE is the result of treatment or the disease itself, cannot be determined.

On the other hand, this study is the first to investigate the incidence of SCFE in Korea using a nationwide cohort database containing information on nearly the entire population of the country. Since 2000, approximately 97.0% of the population in South Korea has been required to enroll in the National Health Insurance Service program. Considering that almost all cases of SCFE that have occurred in Korea are included in the HIRA database, we believe that our study findings are representative of the overall incidence of SCFE in Korea.

## Conclusion

We have investigated the epidemiologic and demographic characteristics of patients with SCFE in South Korea using a nationwide population-based database. The annual incidence rate of SCFE in this country has increased significantly since 2009. Noting the results of this

study, SCFE should be considered as a differential diagnosis in adolescent patients with hip pain.

## Acknowledgments

This study used HIRA research data (M20210107929) made by Health Insurance Review & Assessment Service (HIRA). The views expressed are those of the author(s) and not necessarily those of the HIRA and the MOHW.

## Author Contributions

**Conceptualization:** Moon Seok Park, Ki Hyuk Sung.

**Data curation:** Nak Tscheol Kim, Eunjeong Ji.

**Formal analysis:** Jae Jung Min, Eunjeong Ji.

**Methodology:** Ki Hyuk Sung.

**Supervision:** Moon Seok Park, Ki Hyuk Sung.

**Writing – original draft:** Nak Tscheol Kim.

**Writing – review & editing:** Jae Jung Min.

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
