## [Decision Letter · Decision Letter 0]

25 May 2022

PONE-D-21-25828

Epidemiology and demographics of slipped capital femoral epiphysis in South Korea: a nationwide cohort study

PLOS ONE

Dear Dr. Sung,

Thank you for submitting your manuscript to PLOS ONE. After careful consideration, we feel that it has merit but does not fully meet PLOS ONE’s publication criteria as it currently stands. Therefore, we invite you to submit a revised version of the manuscript that addresses the points raised during the review process.

Your manuscript has been assessed by an expert reviewer, whose comments are appended below. The reviewer has raised several questions about aspects of the methodology and discussion which you should address carefully in your revised manuscript.

Please note that we have only been able to secure a single reviewer to assess your manuscript. We are issuing a decision on your manuscript at this point to prevent further delays in the evaluation of your manuscript. Please be aware that the editor who handles your revised manuscript might find it necessary to invite additional reviewers to assess this work once the revised manuscript is submitted. However, we will aim to proceed on the basis of this single review if possible. 

We look forward to receiving your revised manuscript.

Kind regards,

Joseph Donlan

Editorial Office

PLOS ONE

Reviewers' comments:

Reviewer's Responses to Questions

**Comments to the Author**

1. Is the manuscript technically sound, and do the data support the conclusions?

Reviewer #1: Yes

2. Has the statistical analysis been performed appropriately and rigorously? 

Reviewer #1: Yes

3. Have the authors made all data underlying the findings in their manuscript fully available?

Reviewer #1: Yes

4. Is the manuscript presented in an intelligible fashion and written in standard English?

Reviewer #1: Yes

5. Review Comments to the Author

Reviewer #1: A very nice study. I have some comments that I believe will strengthen the study.

Line 42 – SCFE is not truly a separation except in the unstable type. The more common stable type is just a shift in position of the epiphysis to the metaphysis. Please reword this.

Line 47 – causing the epiphysis to slip relative to the metaphysis – it is not the femoral that head that is slipping, as the head included both the epiphysis and the proximal metaphysic.

Methods

Are all Korean patients in the HIRA database, or only those with government insurance. I am sorry I don’t know how Korean insurance works – are there private insurance companies as well as a national health insurance. I think American readers would want to know how this works.

Also, a general overall question – how accurate is the HIRA database – are there some cases that would be missed, or secondary information for a SCFE case (eg endocrine disorders, etc). I think this should be mentioned in the Material and Methods section. Have accuracy studies been done?

Results

Line 111 instead of using the word comorbidity, I would use disease, as done in the same sentence with pituitary disease.

Line 116-117 – can this be further explained? Were the 15 cases of same day surgery due to simultaneous bilateral SCFE, and the 64 for patients that developed a 2nd sequential SCFE later on?

Line 119 – Was the femoral osteotomy the treatment for the SCFE or for the AVN; same question for the pelvic osteotomy and total hip arthroplasty.

Do we know why AVN developed? Was it because they were unstable SCFEs or was it due to treatment of the SCFE (eg modified Dunn procedure, etc)? Please amplify. If not known because it wasn’t in the data base please state so.

Line 143 – delete “detectable” it was obviously an increase!

Line 154 and Figure 4 – are these data for Korea? I assume so, but please state, and include in the legend for Figure 4.

Lines 176-178 – how can it be assumed the 15 bilateral cases were for prophylactic fixation? If it is not stated in the HIRA database just state that – certainly simultaneous bilateral SCFEs exist, and in the US account for about 50% of bilateral cases.

Discussion:

Lines 191-192 - this answers one of my previous questions – but please state it earlier so that the reader is now wondering until the last paragraph.

Table 1: What is meant by estrogen therapy? Why was it given? For what disease? As is known, estrogen causes physeal closure and tends to be protective against SCFE (thus why less common in adolescent girls than boys). Thus can we be sure that the estrogen therapy was causative for the SCFE or just an association?

Figures.

Figure 1 – can be easily deleted as the information is given in the Results.

Figures 2 and 3 are nice.

6. PLOS authors have the option to publish the peer review history of their article (what does this mean?). If published, this will include your full peer review and any attached files.

Reviewer #1: No

---

## [Author Response · Author response to Decision Letter 0]

13 Jul 2022

Thank you for your time and effort in reviewing this paper. We have made the appropriate changes to the manuscript.

Line 42 – SCFE is not truly a separation except in the unstable type. The more common stable type is just a shift in position of the epiphysis to the metaphysis. Please reword this.

Thank you for the comment. We have reworded this definition in the manuscript as follows: 

“Slipped capital femoral epiphysis (SCFE) is associated with many complications and deformities secondary to a shift in position of the epiphysis in relation to the metaphysis of the proximal femur.”

“When combined with increased body weight and retroversion of the femoral head, the loads may exceed the ultimate yield point, causing a shift in position of the epiphysis in relation to the metaphysis of the femoral head. In severe cases, which are classified as unstable, a true separation may occur.” 

Line 47 – causing the epiphysis to slip relative to the metaphysis – it is not the femoral that head that is slipping, as the head included both the epiphysis and the proximal metaphysis.

Thank you for your comment. We have made the following changes: 

“When combined with increased body weight and retroversion of the femoral head, the loads may exceed the ultimate yield point, causing a shift in position of the epiphysis in relation to the metaphysis of the femoral head. In severe cases, which are classified as unstable, a true separation may occur.”

Methods

Are all Korean patients in the HIRA database, or only those with government insurance. I am sorry I don’t know how Korean insurance works – are there private insurance companies as well as a national health insurance. I think American readers would want to know how this works.

Thank you for this comment. The Korean healthcare is a national health insurance, and all medical costs are covered by the national insurance system. It would be rational to assume that all Korean patients are in the HIRA database. We have clarified this point in the manuscript as following:

“All Korean patients are enlisted in HIRA, as the Korean healthcare is a national health insurance.”

Also, a general overall question – how accurate is the HIRA database – are there some cases that would be missed, or secondary information for a SCFE case (eg endocrine disorders, etc). I think this should be mentioned in the Material and Methods section. Have accuracy studies been done?

All patients in the HIRA database are enlisted in accordance with diagnoses made by their doctors, and the records are stored electronically. Diseases that do not require a specialist opinion for diagnosis, such as the common cold, may be less accurately reflected in the database. On the other hand, diseases such as SCFE that require a pediatric orthopedic board-certified doctor for diagnosis are very accurately reflected in the HIRA database. Although a separate accuracy study has not been conducted, we believe that the process of patient registration guarantees the accuracy of the HIRA database regarding the current study on SCFE. 

We have clarified this point in the Material and Methods section as follows: 

“Nearly all information on patients and their medical records can be obtained from this database, which has been used in previous epidemiology studies. Accuracy of the database regarding SCFE is guaranteed as patients diagnosed with SCFE are registered in HIRA upon diagnosis made by pediatric orthopedic specialists.” 

Results

Line 111 instead of using the word comorbidity, I would use disease, as done in the same sentence with pituitary disease.

Thank you for your comment. We have made the relevant changes. 

Line 116-117 – can this be further explained? Were the 15 cases of same day surgery due to simultaneous bilateral SCFE, and the 64 for patients that developed a 2nd sequential SCFE later on?

This is correct. Of the 79 patients who were recorded in the HIRA database as having undergone bilateral hip operations, 15 have undergone bilateral in-situ pinning on the same day. Bilateral in-situ pinning was done with some time interval in between in the remaining 64 patients, possibly due to second sequential SCFE. However, whether bilateral in-situ pinning was done simultaneously because of bilateral SCFE or for prophylactic purposes cannot be determined with data extracted from the HIRA database. We have clarified this in the discussion and included it as a limitation:

Discussion:

“In our study, 79 patients (13.5%) underwent bilateral hip surgery. In 15 cases, bilateral surgery was performed on the same day; however, whether these procedures were done for prophylactic purposes or for treatment of bilateral SCFE cannot be ascertained given the nature of the HIRA database.” 

Limitations:

“This study has some limitations, which stem mainly from its retrospective cohort design. First, we could not account for misclassifications, such as incorrect billing or wrong diagnostic codes. However, only patients with a diagnosis of SCFE who underwent related surgery within 90 days were included in the study, which we believe minimized the likelihood of the initial diagnosis being incorrect. Second, we could not test for a correlation between the incidence of SCFE and obesity because the HIRA database does not include information on BMI. Third, it is difficult to discern the reason for simultaneous bilateral hip surgeries with the HIRA database. We cannot know whether simultaneous bilateral hip surgeries were done for prophylactic purposes or for treatment of bilateral SCFE, for the database lacks information on laterality.”

Line 119 – Was the femoral osteotomy the treatment for the SCFE or for the AVN; same question for the pelvic osteotomy and total hip arthroplasty.

Femoral, pelvic osteotomy and total hip arthroplasty were done for treatment of AVN. We have further clarified as follows:

“Avascular necrosis occurred at a mean of 26.4 months after the initial surgery for SCFE in 25 patients. Twelve patients with AVN have undergone femoral osteotomy, with one case each of pelvic osteotomy and total hip arthroplasty.”

Do we know why AVN developed? Was it because they were unstable SCFEs or was it due to treatment of the SCFE (eg modified Dunn procedure, etc)? Please amplify. If not known because it wasn’t in the data base please state so.

The etiology of AVN after SCFE cannot be discerned with information from the HIRA database. We have added this to the limitations:

“This study has some limitations, which stem mainly from its retrospective cohort design. First, we could not account for misclassifications, such as incorrect billing or wrong diagnostic codes. However, only patients with a diagnosis of SCFE who underwent related surgery within 90 days were included in the study, which we believe minimized the likelihood of the initial diagnosis being incorrect. Second, we could not test for a correlation between the incidence of SCFE and obesity because the HIRA database does not include information on BMI. Third, it is difficult to discern the reason for simultaneous bilateral hip surgeries with the HIRA database. We cannot know whether simultaneous bilateral hip surgeries were done for prophylactic purposes or for treatment of bilateral SCFE, for the database lacks information on laterality. Fourth, the type of SCFE cannot be distinguished with the HIRA database. Therefore, whether AVN after SCFE is the result of treatment or the disease itself cannot be determined.”

Line 143 – delete “detectable” it was obviously an increase!

We deleted the word. Thank you for your comment. 

Line 154 and Figure 4 – are these data for Korea? I assume so, but please state, and include in the legend for Figure 4.

We clarified the legend to state that the data is from a Korean population: 

“Figure 4. Annual increase in obesity rate based on body mass index (BMI) since 2009 in the Korean population aged 9–14 years.”

Lines 176-178 – how can it be assumed the 15 bilateral cases were for prophylactic fixation? If it is not stated in the HIRA database just state that – certainly simultaneous bilateral SCFEs exist, and in the US account for about 50% of bilateral cases.

Thank you for the comment. With the HIRA database, the reason for simultaneous bilateral cases cannot be discerned. We have made the relevant corrections as follows: 

“In our study, 79 patients (13.5%) underwent bilateral hip surgery. In 15 cases, bilateral surgery was performed on the same day; however, whether these procedures were done for prophylactic purposes or for treatment of bilateral SCFE cannot be ascertained given the innate nature of the HIRA database.” 

We have also included this shortcoming in the limitations section:

“This study has some limitations, which stem mainly from its retrospective cohort design. First, we could not account for misclassifications, such as incorrect billing or wrong diagnostic codes. However, only patients with a diagnosis of SCFE who underwent related surgery within 90 days were included in the study, which we believe minimized the likelihood of the initial diagnosis being incorrect. Second, we could not test for a correlation between the incidence of SCFE and obesity because the HIRA database does not include information on BMI. Third, it is difficult to discern the reason for simultaneous bilateral hip surgeries with the HIRA database. We cannot know whether simultaneous bilateral hip surgeries were done for prophylactic purposes or for treatment of bilateral SCFE, for the database lacks information on laterality.”

Discussion:

Lines 191-192 - this answers one of my previous questions – but please state it earlier so that the reader is now wondering until the last paragraph.

Thank you for your comment. We have clarified the figure legend to more clearly state that this is an epidemiologic study based on the Korean population. 

Table 1: What is meant by estrogen therapy? Why was it given? For what disease? As is known, estrogen causes physeal closure and tends to be protective against SCFE (thus why less common in adolescent girls than boys). Thus can we be sure that the estrogen therapy was causative for the SCFE or just an association?

In cases of delayed puberty, sex hormone therapies can be used for treatment. In case of pituitary insufficiencies, when there are deficiencies in excretion of both growth hormone and sex hormones, estrogen and growth hormone can be administered simultaneously for treatment of short stature and delayed puberty. We assume that the two cases of estrogen therapy are those who were simultaneously administered with growth hormone. Therefore, we agree with your comment that the incidence of SCFE in these two cases are the result of co-administered growth hormone rather than estrogen therapy itself. We have deleted this column from the table.

Table 1. Comorbidities and History of hormone and radiation therapy in SCFE patients

 Total SCFE patients

(N=586) MALE

(N=429) FEMALE

(N=157)

Comorbidity 

Pituitary gland disease 29 (4.95) 11 (2.56) 18 (11.46)

Thyroid disease 60 (10.24) 30 (6.99) 30 (19.11)

Parathyoid disease 12 (2.05) 8 (1.86) 4 (2.55)

Hypogonadism 16 (2.73) 12 (2.80) 4 (2.55)

Turner syndrome 1 (0.17) 0 (0.00) 1 (0.64)

Scurvy 1 (0.17) 1 (0.23) 0 (0.00)

History of hormone therapy 　 　 　

Growth hormone 10 (1.71) 4 (0.93) 6 (3.82)

History of radiation therapy 11 (1.88) 5 (1.17) 6 (3.82)

Figures.

Figure 1 – can be easily deleted as the information is given in the Results.

Figures 2 and 3 are nice.

Thank you for your comment. We will remove Figure 1 to reduce redundancy.

---

## [Decision Letter · Decision Letter 1]

7 Sep 2022

PONE-D-21-25828R1Epidemiology and demographics of slipped capital femoral epiphysis in South Korea: a nationwide cohort studyPLOS ONE

Dear Dr. Sung,

Thank you for submitting your manuscript to PLOS ONE. After careful consideration, we feel that it has merit but does not fully meet PLOS ONE’s publication criteria as it currently stands. Therefore, we invite you to submit a revised version of the manuscript that addresses the points raised during the review process.

The manuscript has been evaluated by a further two reviewers, and their comments are available below.

The reviewers raised a number of major concerns that need attention. Could you please revise the manuscript to carefully address the concerns raised?

We look forward to receiving your revised manuscript.

Kind regards,

Jamie Royle

Staff Editor

PLOS ONE

Journal Requirements:

Reviewers' comments:

Reviewer's Responses to Questions

**Comments to the Author**

1. If the authors have adequately addressed your comments raised in a previous round of review and you feel that this manuscript is now acceptable for publication, you may indicate that here to bypass the “Comments to the Author” section, enter your conflict of interest statement in the “Confidential to Editor” section, and submit your "Accept" recommendation.

Reviewer #2: All comments have been addressed

Reviewer #3: All comments have been addressed

2. Is the manuscript technically sound, and do the data support the conclusions?

Reviewer #2: Yes

Reviewer #3: Yes

3. Has the statistical analysis been performed appropriately and rigorously? 

Reviewer #2: Yes

Reviewer #3: Yes

4. Have the authors made all data underlying the findings in their manuscript fully available?

Reviewer #2: Yes

Reviewer #3: Yes

5. Is the manuscript presented in an intelligible fashion and written in standard English?

Reviewer #2: Yes

Reviewer #3: Yes

6. Review Comments to the Author

Reviewer #2: All queries and comments have adequately been attended to.

It reads well and is scientifically sound.

Reviewer #3: Authors answered all previous reviewer's concerns.

There are still, however, some points that deserve further clarification:

1) Authors should provide rate of Stable vs Unstable SCFE, and repeat analysis separately (rate of complication is different between the two groups). This separation should also be reflected in their analysis according to age. Comparison should then be made.

2) Authors did not mention 'pistol grip deformity'.

3) How AVN was assessed?

4) Patients flowchart is missing (and it should include the distinction between ambulatory and non-ambulatory patients).

5) More discussion about changing trend in obesity (it looks the rate of younger patients with obesity is increasing and getting closer to older patients). Is the rate of SCFE reflected by this trend? Is the proportion of SCFE the same at all ages (year by year, please, for this particular aspect. This should be discussed +++. Please implement Discussion.

6) What is the importance of your findings in clinical practice? To be discussed.

7) Idiopathic SCFE and SCFE with underlying pathology/secondary SCFE should be analyzed separately in term of epidemiology and demographic, and be compared. Please revise.

7. PLOS authors have the option to publish the peer review history of their article (what does this mean?). If published, this will include your full peer review and any attached files.

Reviewer #2: No

Reviewer #3: No

---

## [Author Response · Author response to Decision Letter 1]

15 Oct 2022

1) Authors should provide rate of Stable vs Unstable SCFE, and repeat analysis separately (rate of complication is different between the two groups). This separation should also be reflected in their analysis according to age. Comparison should then be made.

Thank you for your comments. As the raw data we used for the analysis in this study are HIRA data, a separate analysis between stable and unstable SCFE cannot be performed due to the inherent nature of the HIRA database. We will include this point in the limitations section as follows.

“Fourth, the type of SCFE cannot be distinguished with the HIRA database. Therefore, the rate of stable and unstable SCFE, as well as whether AVN after SCFE is the result of treatment or the disease itself, cannot be determined.”

2) Authors did not mention 'pistol grip deformity'.

In the HIRA database, the diagnosis was based on ICD-10 terminology. In ICD-10, pistol grip deformity, also known as cam-type femoroacetabular impingement (FAI), is registered as femoroacetabular impingement. Therefore, we used the term FAI instead of pistol grip deformity. We will add a pistol grip deformity in the appropriate section of FAI.

“Femoroacetabular impingement (pistol grip deformity) occurred at an average of 45.6 months after the initial surgery in 13 patients and chondrolysis at an average of 9.3 months in two patients.” 

3) How AVN was assessed?

As mentioned in the text, the AVN was assessed according to its diagnostic code in the HIRA database. 

“Avascular necrosis, femoroacetabular impingement, and chondrolysis were identified by their diagnostic codes alone.”

4) Patient flowchart is missing (and it should include the distinction between ambulatory and non-ambulatory patients).

Thank you for your comments. Because the data we used were retrieved from the HIRA database, it is not feasible to create a patient flow chart. For the same reason, distinguishing between ambulatory and non-ambulatory patients is also not feasible. 

5) More discussion about changing trend in obesity (it looks the rate of younger patients with obesity is increasing and getting closer to older patients). Is the rate of SCFE reflected by this trend? Is the proportion of SCFE the same at all ages (year by year, please, for this particular aspect. This should be discussed +++. Please implement Discussion.

Thank you for your comments. The HIRA data did not reflect body mass index (BMI). Therefore, we retrieved the data for BMI and analyzed trends according to age. We postulate that the increase in the incidence of SCFE is related to an increasing trend in childhood obesity. We have included this shortcoming in the Limitations section.

“We could not test for a direct correlation between the incidence of SCFE and obesity because BMI was not included in the HIRA data. However, the obesity rate among Korean children and adolescents has been steadily increasing (Fig. 3) [25]. We assume that this steady increase in the rate of obesity, which is already a known risk factor for SCFE, may be an important environmental factor that influences the increase in the incidence of SCFE in Korea.”

6) What is the importance of your findings in clinical practice? To be discussed.

With the increase in childhood obesity, the incidence of SCFE also seems to increase. Although the prevalence of SCFE is lower in Korea than in Western countries, due to its increased incidence, SCFE should not be excluded from the differential diagnosis when an adolescent patient with hip pain is brought to the clinic. We have included this point in the conclusion as follows: 

“We could not test for a direct correlation between the incidence of SCFE and obesity because BMI was not included in the HIRA data. However, the obesity rate among Korean children and adolescents is steadily increasing (Fig. 3) [25]. We assume that this steady increase in the rate of obesity, which is already a known risk factor for SCFE, may be an important environmental factor influencing the increased incidence of SCFE in Korea. 

In the clinical setting, SCFE should be included in the differential diagnosis of adolescent patients with hip pain.” 

7) Idiopathic SCFE and SCFE with underlying pathology/secondary SCFE should be analyzed separately in terms of epidemiology and demographics and compared. Please revise.

Thank you for your comment. As described in the Results section, 95 (16.2%) of the 586 patients had endocrine comorbidities, a history of hormone therapy, or received radiation therapy. However, the access to the raw data needed to analyze and compare idiopathic and secondary SCFE is closed and it will take at least 1 year to regain access to the database. Please understand that for such a realistic reason, the analysis is difficult under current circumstances.

---

## [Decision Letter · Decision Letter 2]

25 Nov 2022

PONE-D-21-25828R2Epidemiology and demographics of slipped capital femoral epiphysis in South Korea: a nationwide cohort studyPLOS ONE

Dear Dr. Sung,

Thank you for submitting your manuscript to PLOS ONE. After careful consideration, we feel that it has merit but does not fully meet PLOS ONE’s publication criteria as it currently stands. Therefore, we invite you to submit a revised version of the manuscript that addresses the points raised during the review process.

We look forward to receiving your revised manuscript.

Kind regards,

Faizan Iqbal

Academic Editor

PLOS ONE

Journal Requirements:

Reviewers' comments:

Reviewer's Responses to Questions

**Comments to the Author**

1. If the authors have adequately addressed your comments raised in a previous round of review and you feel that this manuscript is now acceptable for publication, you may indicate that here to bypass the “Comments to the Author” section, enter your conflict of interest statement in the “Confidential to Editor” section, and submit your "Accept" recommendation.

Reviewer #1: All comments have been addressed

Reviewer #3: All comments have been addressed

2. Is the manuscript technically sound, and do the data support the conclusions?

Reviewer #1: Yes

Reviewer #3: Yes

3. Has the statistical analysis been performed appropriately and rigorously? 

Reviewer #1: Yes

Reviewer #3: Yes

4. Have the authors made all data underlying the findings in their manuscript fully available?

Reviewer #1: No

Reviewer #3: Yes

5. Is the manuscript presented in an intelligible fashion and written in standard English?

Reviewer #1: Yes

Reviewer #3: Yes

6. Review Comments to the Author

Reviewer #1: All my comments have been answered. This is an excellent study and will be an addition to the SCFE epidemiology literature.

Reviewer #3: Thank you for making requested changes; the manuscript reads much better.

One additional comment:

Authors should add a Table summarizing similar studies assessing the epidemiology of SCFE, and its evolution over time, in other parts of the world, and compare percentages and type of study with yours. This should be presented in Table format.

7. PLOS authors have the option to publish the peer review history of their article (what does this mean?). If published, this will include your full peer review and any attached files.

Reviewer #1: No

Reviewer #3: No

---

## [Author Response · Author response to Decision Letter 2]

6 Dec 2022

Thank you for your comment. We have added “table 3” to enlist and summarize epidemiology of SCFE in other countries and compared it to the results of our study. 

*Table attached*

---

## [Editor Report · Decision Letter 3]

21 Feb 2023

PONE-D-21-25828R3

Epidemiology and demographics of slipped capital femoral epiphysis in South Korea: a nationwide cohort study

PLOS ONE

Dear Dr. Sung,

Thank you for submitting your manuscript to PLOS ONE. After careful consideration, we feel that it has merit but does not fully meet PLOS ONE’s publication criteria as it currently stands. Therefore, we invite you to submit a revised version of the manuscript that addresses the points raised during the review process.

We look forward to receiving your revised manuscript.

Kind regards,

Faizan Iqbal

Academic Editor

PLOS ONE

Journal Requirements:

Editor's comments:

Authors should add a Table summarizing similar studies assessing the epidemiology of SCFE, and its evolution over time, in other parts of the world, and compare percentages and type of study with yours. This should be presented in Table format.

---

## [Author Response · Author response to Decision Letter 3]

24 Feb 2023

We thank the editors and the reviewers for their time and effort in reviewing this study. We have revised “table 3” according to date of study to enlist and summarize epidemiology of SCFE in other countries and compared it to the results of our study, which is stated at the last row of the table.

---

## [Editor Report · Decision Letter 4]

3 Mar 2023

Epidemiology and demographics of slipped capital femoral epiphysis in South Korea: a nationwide cohort study

PONE-D-21-25828R4

Dear Dr. Sung,

We’re pleased to inform you that your manuscript has been judged scientifically suitable for publication and will be formally accepted for publication once it meets all outstanding technical requirements.

Kind regards,

Faizan Iqbal

Academic Editor

PLOS ONE
---

## [Editor Report · Acceptance letter]

22 Mar 2023

PONE-D-21-25828R4 

Epidemiology and demographics of slipped capital femoral epiphysis in South Korea: a nationwide cohort study 

Dear Dr. Sung:

I'm pleased to inform you that your manuscript has been deemed suitable for publication in PLOS ONE. Congratulations! Your manuscript is now with our production department. 

Kind regards, 

on behalf of

Dr. Faizan Iqbal 

Academic Editor

PLOS ONE